# Assessment of Thermal Comfort and Air Quality of Room Conditions by Impinging Jet Ventilation Integrated with Ductless Personalized Ventilation

Bin Yang [1,2,*], Pengju Liu [1], Yihang Liu [1], Dacheng Jin [1] and Faming Wang [3]

1   School of Building Services Science and Engineering, Xi'an University of Architecture and Technology, Xi'an 710055, China
2   School of Energy and Safety Engineering, Tianjin Chengjian University, Tianjin 300384, China
3   Department of Biosystems, KU Leuven, 3001 Leuven, Belgium
*   Correspondence: yangbin@xauat.edu.cn; Tel.: +86-022-23085102

**Abstract:** Advanced ventilation methods are responsible for creating an appropriate temperature environment with satisfactory inhaled air quality. The ductless personalized ventilation system integrated with impinging jet ventilation shows the good ventilation performance. In order to investigate the effect of using such an integrated system on thermal comfort and air quality improvement. Twenty subjects participated in a chamber test at 25 °C, 27 °C, and 29 °C, respectively, with operating DPV devices at three modes (no flow, pre-set flow, and user control flow). Votes on thermal comfort, thermal sensation, thermal acceptability, and perceived air quality were collected from the them. The results showed that overall thermal sensation votes with DPV running at the user control flow mode were close to neutral (0.1, 0.4, and 0.5, respectively, at 25 °C, 27 °C, and 29 °C). Thermal comfort and perceived air quality were improved at all three temperatures studied in the user control DPV flow mode, with 90% of occupants reporting that the thermal environments were acceptable. An integrated system of this type could raise the acceptable HVAC temperature setpoint to 29 °C, resulting in an average energy savings of 34% over the neutral condition at 25 °C. Hence, occupants are advised to use the DPV's user-control mode. Lastly, it is concluded that the integrated system could greatly improve thermal comfort, perceived air quality, and save HVAC energy, despite some issues with dry eyes at 29 °C.

**Keywords:** impinging jet ventilation; ductless personalized ventilation; thermal comfort; thermal sensation; perceived air quality

## 1. Introduction

One of the primary goals of the building is to provide a comfortable environment that does not jeopardize the occupants' health or work performance [1]. In practice, building and construction industries account for 36% of global final energy consumption and over 40% of total direct and indirect $CO_2$ emissions [2]. Heating, ventilation, and air conditioning (HVAC) systems account for a significant portion of energy consumption in modern buildings for achieving a desirable indoor climate [3], e.g., in tropical climates, the energy consumption by an HVAC system may exceed 50% of the total energy consumption of a building [4]. In China, rules have been tightened to reduce carbon emissions and reach carbon neutrality by 2060 [5]. As a result, HVAC systems must provide both optimal indoor conditions and energy efficiency [3]. In HVAC, ventilation is critical for delivering adequate indoor air quality (IAQ) while also being responsible for energy consumption in buildings [6]. The lower section of a room (usually less than 1.8 m) is the primary zone for occupant activities; therefore, an efficient ventilation system in the occupied region is required. In practical applications, the standard mixing ventilation system (MV) offsets the full space loads, resulting in poor ventilation efficiency and lower energy efficiency [7].

Displacement ventilation (DV), which only offsets occupied zone loads, has been shown to improve IAQ [8], thermal comfort [9] and be more energy efficient than MV [10]. Design criteria for DV systems are relatively mature, and there are numerous successful engineering projects available [7]. As a result, in many scenes, DV has been preferred over MV.

Even for one person, the favorite setting might fluctuate from day to day and even within the same day [11]. Because it may be flexibly modified by the user, personalized ventilation (PV) is an appropriate solution for satisfying an occupant's micro-environment desire [12]. A PV system immediately feeds clean air to the occupant's breathing zone, providing for good inhaled air quality. PV is frequently used in conjunction with an ambient ventilation system, such as MV and DV, to provide a preferred climate while remaining energy-efficient [13,14]. Nonetheless, the architectural requirements and additional duct installation would limit the application of a PV system. So, a fresh sort of individualized ventilation, the ductless personalized ventilation (DPV) system, was recently proposed. As a self-contained system, DPV does not require a duct, making it more conducive to indoor aesthetics and room layout flexibility than PV [15,16]. The DPV is typically used in conjunction with the DV system because it may absorb the cool air supplied by the DV near the floor and convey it to the occupant's breathing zone [11,17].

Several studies of DPV's performance were conducted when DV served as the background system. Halvoňová et al. [18] investigated the DPV system disturbance caused by a person walking to the displacement diffuser, as well as how the DPV system was influenced by the workstation layout [19]. They reported that neither the workstation arrangement nor the walking person had any discernible negative impact on DPV performance. Alsaad et al. used the CFD method to assess thermal comfort and air quality of an indoor environment equipped with a ductless personalized ventilation system [20]. Results showed that both thermal comfort and indoor air quality had improved. They also compared the inhaled air quality performance of a desk fan and a DPV system [21]. The results demonstrated that the DPV system performed superiorly in removing exhaled contaminants. Katramiz et al. reported that when combined with DV, DPV protects children from potential cross-contamination in classrooms [22].

Although studies have shown that when used in conjunction with a DV system, DPV improved inhaled air quality [17], energy efficiency [23], and thermal comfort [20], limitations must be noted. This is because the background system, the DV, operated with a low supply momentum [24]. In a DV system, the low momentum would be consumed by the plume when passing the heat source, and poor ventilation efficiency would result in some regions of the room [25,26]. Due to this limitation, the DPV system is used in a small office with low heat density [19,22], e.g., with a heat load of 22.6 W/m$^2$. To maintain the good performance of the DPV system in an office with a typical heat load, an additional system, such as a cooling radiant floor system, must be used to accommodate the heat loads [16].

To benefit from the good performance of DPV in the typical office, a higher performance stratified ventilation system should serve as the background system. To address the shortcomings of DV, impinging jet ventilation (IJV), with high ventilation performance, was developed in Sweden [24,27]. The supplied air jets downward with high momentum on the floor and spreads over a large area in the IJV system, resulting in a balance between momentum in the supply air and buoyancy forces due to heat sources, and sufficient force is achieved to reach long distances [24,28]. Furthermore, the vertical air temperature of IJV is similar to that of DV, implying that cooler air distributes in the bottom section of the occupied zone. As a result, the IJV system, which has a high ventilation efficacy, has been investigated as a possible background ventilation system when using the DPV in our pilot study, as shown in Figure 1 [29]. Findings revealed that when IJV served as the background system, the DPV performed well in terms of providing local cooling even in warm conditions. However, it has not been determined whether DPV + IJV contributes to overall thermal comfort and indoor air quality of personnel, which is the premise of whether IJV can be used as a background system application.

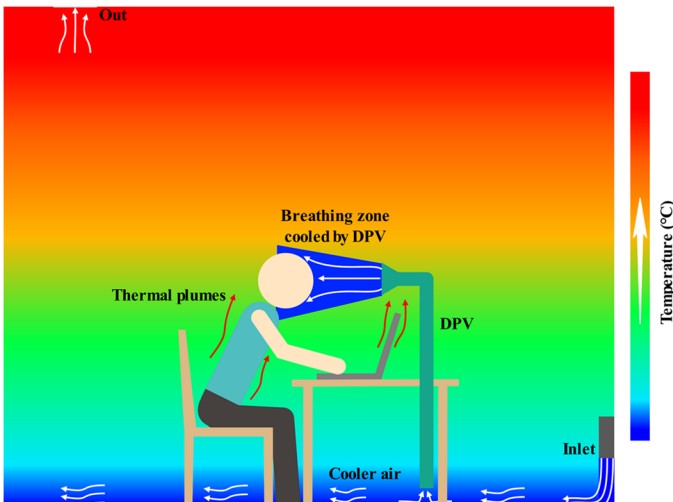

**Figure 1.** Temperature profile under ductless personalized ventilation integrated with impinging jet ventilation.

As a result, the current study looked into whether using ductless personalized ventilation in conjunction with impinging jet ventilation could improve thermal comfort and perceived air quality, particularly in warm conditions. At three indoor temperatures (25 °C, 27 °C, and 29 °C), subjects participated in three DPV modes (no DPV flow, prefixed DPV flow, and user control flow). During the chamber test, subjective votes on the various thermal conditions were collected to assess the impact of using this integrated system on occupant thermal comfort and perceived air quality.

## 2. Materials and Methods

### 2.1. Experiment Setup

The present study was performed in a controlled climate chamber, i.e., Chamber A in Figure 2a, at the Sino-Nordic Research Center for Indoor Environment and Energy in Xi'an, China, during summer. The size of the chamber is 5.4 m (Length) × 5.0 m (Width) × 2.6 m (Height). The chamber setup can be considered as a typical office environment. The chamber can control the indoor thermal environment at a temperature range of 15–40 °C (accuracy: ±0.3 °C), and a relative humidity range of 20–95% (accuracy: ±5%).

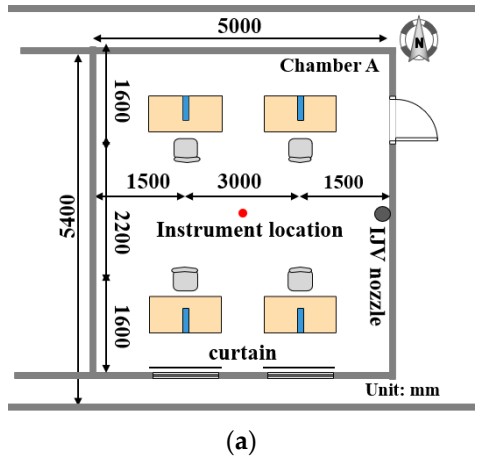

(**a**)

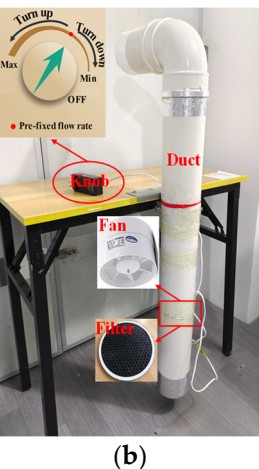

(**b**)

**Figure 2.** Chamber test scene: (**a**) Chamber setup; (**b**) DPV devices.

On the chamber's sidewall, a circular supply terminal (D = 0.25 m) of impinging jet ventilation was installed. The typical office environment included four workstations, resulting in a staff occupied space of 6.75 m$^2$/person. Each workstation had a desk, a

chair, a DPV device that could be adjusted, and a laptop PC. The distance between the table and chair was adjusted before each experiment to achieve a distance ranging from 0.35 m to 0.45 m between the subject's face and the DPV supply air outlet. Figure 2b depicts the details of the DPV device; an axial fan and an activated carbon filter were put in the duct, and insulating material was placed on the inner wall of the air duct to prevent heat transmission through the pipe wall. A knob was also supplied for the user to adjust the DPV device.

### 2.2. Experiment Conditions and Measurement Equipment

The present study was conducted in three different indoor ambient temperatures (measured in the center of the chamber at a height of 1.1 m, i.e., the subject's head level [15,30]) and relative humidity ($45 \pm 5\%$). The impinging jet ventilation system was used to control the indoor temperature, which was set at 25 °C, 27 °C, and 29 °C, respectively. The ventilation systems were run at three different degrees of temperature difference (between supply air temperature and indoor air temperature at 1.1 m above the floor), namely 4 °C, 5 °C, and 6 °C, respectively; additionally, measurable details from the experiments are shown in Table 1.

**Table 1.** Measured parameters during experiments.

| Condition | Temperature at 1.1 m (°C) | Supply Air Temperature (°C) | The Supply Flow Rate of IJV (L/s) | $CO_2$ Concentration at 1.1 m (ppm) |
|---|---|---|---|---|
| 1 | $25.2 \pm 0.3$ | $21.1 \pm 0.1$ | 98 | $572 \pm 23$ |
| 2 | $27.3 \pm 0.2$ | $22.0 \pm 0.1$ | 84 | $590 \pm 30$ |
| 3 | $28.8 \pm 0.2$ | $22.8 \pm 0.2$ | 70 | $628 \pm 28$ |

During the experiment, the ambient temperature and relative humidity were monitored at one-minute intervals by a HOBO data logger (HOBO U12–012, Onset, Bourne, MA, USA; accuracy: $\pm 0.35$ °C); air speed and temperature were measured by the Swema anemometers (Swema 03+, Swema AB, Farsta, Sweden; air speed accuracy: $\pm 0.03$ m/s, temperature accuracy: $\pm 0.3$ °C); $CO_2$ concentration was measured by an RTR−576 data logger (T&D Corporation, Nagano, Japan; accuracy: $\pm 50$ ppm). In addition, wireless skin temperature loggers (iButtons DS1922L, Maxim Integrated, San Jose, CA, USA; resolution: 0.0625 °C, accuracy: $\pm 0.5$ °C) were used to record the subjects' skin temperature at one-minute intervals.

### 2.3. Subjects

In the present study, the software G*Power (G*Power Version 3.1.9.6, Heinrich-Heine-Universität Düsseldorf, Düsseldorf, Germany) was used to confirm the sample size and ensure that there was enough power to detect a statistical difference. Assuming an effect of 0.3, a significant level $\alpha$ of 0.05, and a power of 0.8, thus, seventeen subjects could provide enough power to register a statistical difference of comparable magnitude [31]. Twenty young college students (10 males and 10 females) participated in the chamber test and they were divided into five groups of four people. All subjects have lived in Xi'an for more than two years, and the physical characteristics of the 20 subjects are shown in Table 2. All subjects abstained from alcohol, were in good health with no chronic diseases, and agreed to take part in each research for a fee. Subjects were encouraged to plan their activities according to the schedules, ensuring that they did not drink tea or coffee and did not perform any vigorous activity at least one day before the chamber test.

**Table 2.** Physical characteristic of the 20 participants.

| Gender | Age (yr) | Height (m) | Weight (kg) | Body Mass Index (kg/m²) |
|---|---|---|---|---|
| Males | 23.8 ± 1.7 | 1.76 ± 0.04 | 68.10 ± 8.11 | 21.8 ± 1.87 |
| Females | 22.7 ± 2.0 | 1.61 ± 0.06 | 55.03 ± 4.94 | 21.3 ± 1.75 |
| Total | 20.8 ± 1.8 | 1.69 ± 0.09 | 61.70 ± 9.28 | 21.5 ± 1.83 |

The participants had to wear trousers, short-sleeve shirts, socks, shoes, and underwear. Furthermore, during the trial, subjects were permitted to use computers and surf the internet (estimated metabolic rate is 1.1 mets). During the trial, the volunteers were not allowed to know the specifics of the current experimental environment and were urged not to shift their chairs.

### 2.4. Experiment Procedure and Questionnaires

Subjects were instructed to attend early in order to wear skin temperature sensors. Subjects were allowed to enter the experimental room after providing the relevant information; for example, subjects were not permitted to roam around throughout the chamber test. Before each trial, the indoor ambient parameters stabilized, and each chamber test lasted 140 min (see Figure 3). Subjects were invited to enter the chamber and sit comfortable for the first 20 min for temperature adaptation before performing the thermal evaluation in the remaining formal experiment. The formal trial was divided into four stages, namely, stage 1, lasted 30 min: DPV devices off, stage 2, lasted 40 min: devices on at pre-set flow rate (10 L/s); stage 3, lasted 10 min: rest stage (in this stage, subjects were encouraged to adjust the flow knobs to meet their preferences); and stage 4, lasted 40 min: devices on at the flow rate fixed by users' control.

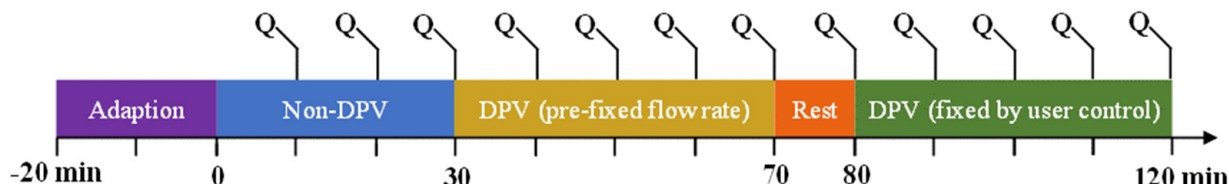

**Figure 3.** Experimental procedure.

Subjects were asked to fill out online questionnaires to report their reactions to the thermal environment during each session. To obtain subjects' thermal states, the ASHRAE 7-point scale was used to assess thermal sensation (−3 cold, −2 cool, −1 slightly cool, 0 neutral, +1 slightly warm, +2 warm, and +3 warm) [32]. In subjective response tests, thermal comfort vote is used to report occupants' satisfaction with the thermal environment, and a 5-point scale was used to assess thermal comfort (−2 uncomfortable, −1 slightly uncomfortable, 0 no feeling, +1 slightly comfortable, and +2 comfortable). A continuous scale from "clearly unacceptable" (−1) to "clearly acceptable " (1) was used to evaluate occupants' thermal acceptability of the thermal environment. Subjects' preferences for the thermal environment were evaluated on a 3-point scale (−1 cooler, 0 no change, and +1 warmer), as did their choice for air movement (−1 lower, 0 no change, and +1 higher).

Perceived air quality is an important factor in evaluating the indoor environment. Occupants will rate the air quality as good if the perceived air is fresh and pleasant [33]. A three-point scale was used to assess the acceptability of perceived air quality (−1 clearly unacceptable, 0 just unacceptable/just acceptable, and +1 clearly acceptable); subjects also reported their feelings about the inhaled air temperature, i.e., the perceived air temperature, using a three-point scale (−1 cool, 0 no feeling, and +1 warm).

Headache, runny nose, dry skin, throat, and dry eyes are all symptoms of the sick building syndrome [34]. So, to evaluate such an integrated system, subjects used a three-

point scale (−1 irritation, 0 moderate irritation, and +1 no irritation) to reporte their responses of the nose, throat, face, and head. The severity of dry eyes was graded on a 5-point scale (−2 dry, −1 slightly dry, 0 neutral, +1 slightly wet, and +2 wet).

### 2.5. Data Analysis

The final 20 min of steady-state data from stages 1, 2, and 4 were evaluated and reported. The Shapiro–Wilk normality test was used to determine whether the obtained data was normal, and normality might be rejected if the *p* value was less than 0.05. Levene's test was used to detect variance homogeneity between data sets. For data with a normal distribution, one-way ANOVA and the paired *t*-test were used. To determine if a significant difference remained in the overall distribution of the non-normally distributed data, the paired Wilcoxon signed-rank test was performed. The Pearson correlation coefficient was used to report the correlation between variables in properly distributed data. Spearman's rank coefficient was used for non-normally distributed data variables. The results were deemed statistically significant at $p < 0.05$ for all test conditions, and the statistical analysis was carried out using SPSS Statistics Version 26.0. (IBM, Chicago, IL, USA).

## 3. Results and Discussion

### 3.1. Skin Temperature

Despite that the face accounts for a very small portion of the body's surface area, face cooling may improve occupant thermal acceptability and thermal comfort [12,35]. The PV system directs cold air at a specific velocity to the breathing zone, lowering skin temperature in PV target areas [36,37].

Figure 4 depicts the local mean temperature at the forehead under all experimental circumstances. At all studied indoor ambient conditions, the local mean temperature at the subject's forehead could be significantly reduced when DPV was delivered at a predetermined flow rate compared to no DPV device ($p < 0.01$, $p < 0.001$, and $p < 0.001$ at 25 °C, 27 °C, and 29 °C, respectively). This demonstrated that when used in conjunction with an IJV system, DPV had the same cooling impact as PV. At the three chosen indoor ambient temperatures, the forehead temperature may be reduced by 1.5, 1.3, and 1.2 °C, respectively. When the DPV devices were used at the fixed flow rate after subjects' control, the forehead temperature could be reduced by 0.8 °C, and no significant difference ($p = 0.053$) in the forehead temperature was discovered when compared to no devices at the indoor ambient temperature of 25 °C. In fact, more subjects were turning down the flow, as observed in Table 3, at 25 °C. Reduced convection causes less heat loss due to convective heat transfer, resulting in higher local temperatures at stage 4 than at stage 1. The opposite, significant differences ($p < 0.001$) persisted with decreases in forehead temperature of 0.9 °C and 1.4 °C under the other two indoor ambient temperatures, respectively. This is because several subjects chose to increase the airflow at warmer conditions.

**Table 3.** User's control (percentage of occupants) on DPV supply devices.

|  | Turn Down | No Change | Turn Up |
|---|---|---|---|
| 25 °C | 70% | 25% | 5% |
| 27 °C | 45% | 15% | 40% |
| 29 °C | 30% | 15% | 55% |

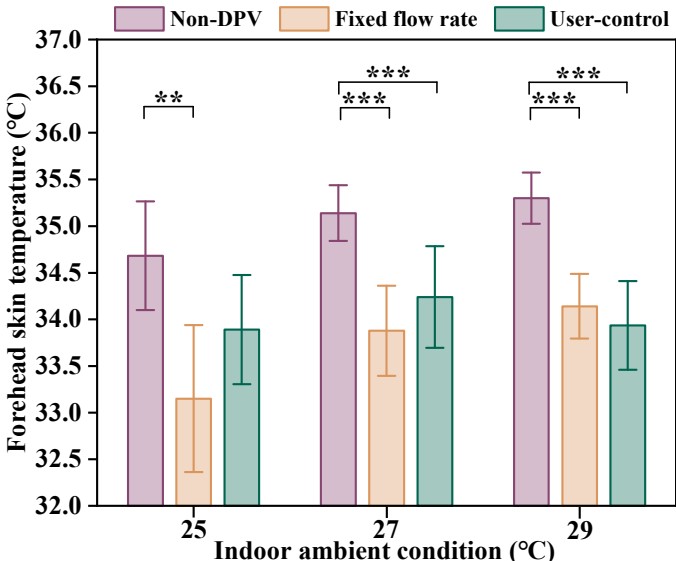

**Figure 4.** Local mean skin temperature at the forehead. Mean value and 95% confidence interval (CI) are shown in the figure. ** $p < 0.01$; ***, $p < 0.001$.

### 3.2. Thermal Sensation Votes

3.2.1. Overall Thermal Sensation (OTS)

Figure 5 depicts the overall thermal sensation (OTS) acquired from all test settings. At the three temperature conditions without DPV, the occupants' mean OTS values were 0 (corresponding to "neutral") at 25 °C, +0.8 (0.4, +1.3) at 27 °C, and +1.1 (+0.7, +1.5) at 29 °C, respectively. When DPV devices with a fixed flow rate were used in stage 2, the OTS was dramatically reduced. The mean value dropped from 0 to −0.2, +0.8 to +0.5, and +1.1 to +0.7 at 25 °C, 27 °C, and 29 °C, respectively. When using user-controlled DPV devices in warmer ambient circumstances, such as 27 °C and 29 °C, OTS was somewhat lower than when using fixed flow rate conditions. In the user control flow mode, the OTS was +0.1, +0.4, and +0.5, respectively, at 25 °C, 27 °C, and 29 °C temperatures.

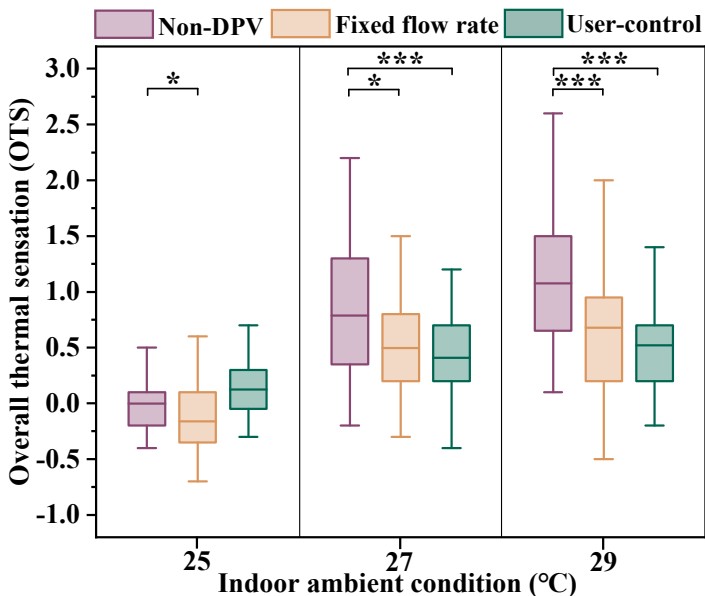

**Figure 5.** Overall thermal sensation. The figure depicts the median, 25% and 75% intervals as well as whiskers for the twenty subjects. *, $p < 0.05$; ***, $p < 0.001$.

### 3.2.2. Local Thermal Sensation

There is a strong linear relationship between the local thermal sensation and the overall sensation [38]. So, some local cooling is used to improve the overall thermal sensation in warm conditions [39]. Figure 6a,b show the subject's local thermal sensibility (LTS) at the head and neck, respectively. When compared to no device settings, the use of DPV devices at pre-set or user-controlled flow rates could dramatically reduce the local hot sensation at the head and neck. The mean value of LTS at the head section could be greatly reduced by 0.65, 0.62, and 0.58 scale units, respectively, by using DPV devices with set flow rates at the three ambient temperatures ($p < 0.001$). Furthermore, LTS was still significantly lower when DPV devices were used in the user-control model compared to no devices ($p < 0.001$), and it remained at the neutral level (0, +0.33, and +0.38, respectively, at indoor ambient temperatures of 25 °C, 27 °C, and 29 °C).

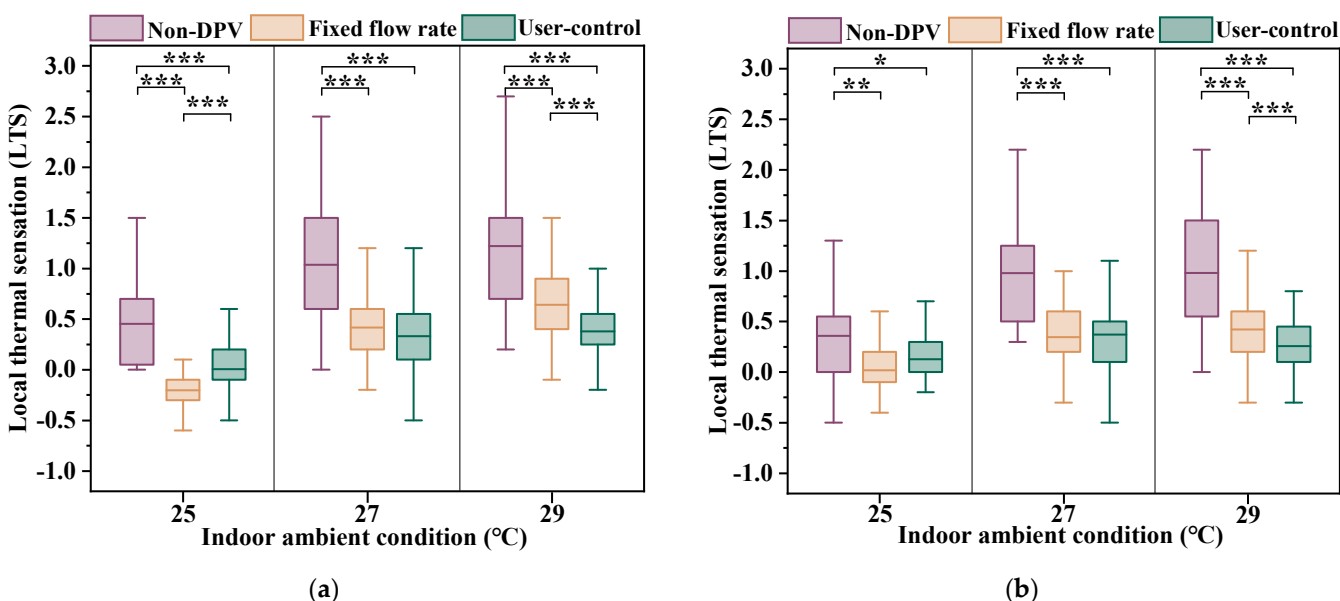

(**a**)         (**b**)

**Figure 6.** Local thermal sensation (LTS): (**a**) at the head part; (**b**) at the neck part. The figure also depicts the median, 25% and 75% intervals and whiskers for the twenty subjects. *, $p < 0.05$; ** $p < 0.01$; ***, $p < 0.001$.

By comparing the LTS of stage 1 to stage 2, the results revealed that the mean value of LTS at the neck section fell by 0.34, 0.63, and 0.56 scale units ($p < 0.01$, $p < 0.001$, and $p < 0.001$, respectively) at the three ambient temperatures with the pre-set DPV flow rates. Furthermore, the mean value of LTS ranged from +0.02 to +0.42 (all values were near to "neutral") for all DPV conditions.

At 25 °C, occupants' local thermal sensation was greater with the user control flow than with the pre-set flow. Local thermal sensation would influence overall thermal sensation [40]. So, while the overall thermal sensation was neutral, OTS was slightly higher with the user control flow mode than with the pre-set flow. It is also worth noting that when the DPV flow was turned off, the OTS with control flow was slightly higher, though the difference was not statistically significant. One possible explanation is that the pre-set flow is inappropriate for them, which affects their judgement of thermal sensation voting in the user control situation. Section 3.3.3 will go over the occupants' choice of flow.

### 3.3. Thermal Comfort

#### 3.3.1. Thermal Comfort Votes

Figure 7 depicts the overall thermal comfort votes (TCVs). In comparison to no devices, the results demonstrated that participants' TCVs could be greatly enhanced when they controlled the DPV devices' providing flow rates. However, at 25 °C indoor temperature,

TCVs reduced marginally when DPV devices were used at the pre-set flow rate and dramatically rose after users modified the devices ($p = 0.01$). Unlike at 25 °C, TCVs of each trial stage (excluding stage 3, which was not considered) improved over the previous stage. TCVs improved more significantly compared to the previous stage. It should be noted that the difference in TCVs between no device and user-control situations was more significant in warmer ambient circumstances ($p = 0.04$ at 25 °C, $p = 0.03$ at 27 °C, and $p < 0.001$ at 29 °C). The results also demonstrated that devices that provide local cooling at the face improved not only the local thermal experience, but also the overall thermal comfort [39,41].

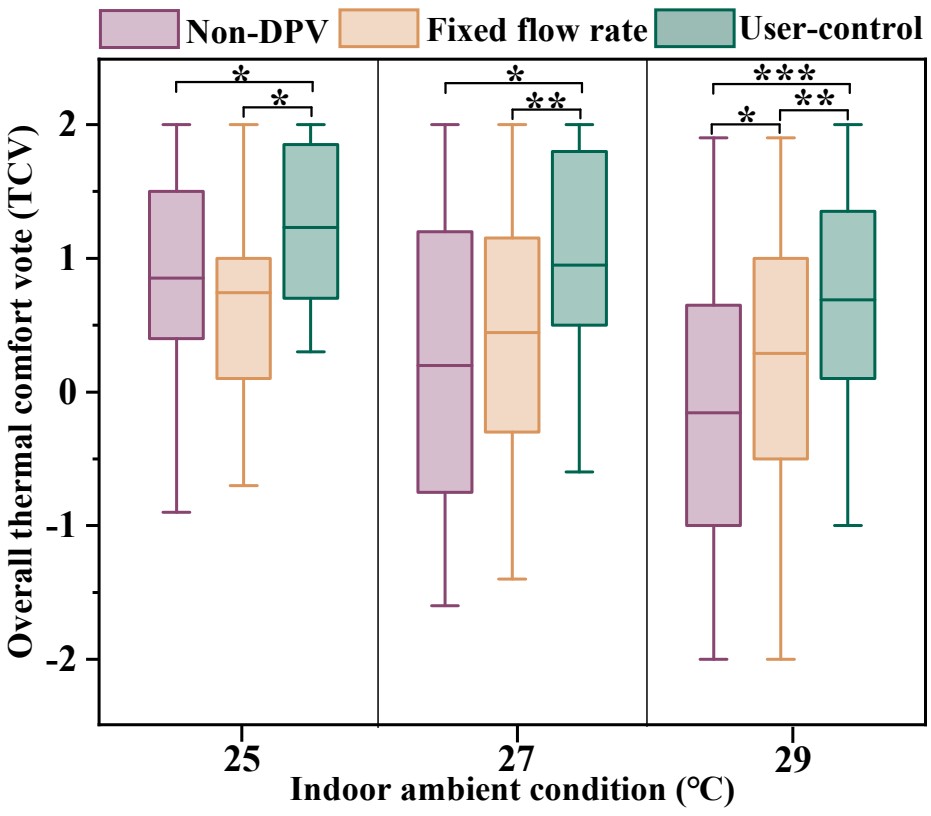

**Figure 7.** Overall thermal comfort votes. The figure also depicts the median, 25% and 75% intervals and whiskers for the twenty subjects. *, $p < 0.05$; ** $p < 0.01$; ***, $p < 0.001$.

### 3.3.2. Thermal Acceptability and Thermal Preference

Figure 8a shows the thermal preference of subjects, i.e., how they desire to adjust the current temperature environment (warmer, cooler, or no change). At the three ambient temperatures, the subjects' readiness to adjust the thermal environment fell significantly when DPV devices were used against no devices, especially when the devices were used under the users' control (25% to 10%, 70% to 40%, and 85% to 35%). This is due to the adjustable DPV airflow meeting some of the occupants' thermal comfort requirements. Figure 8 depicts the acceptability of the thermal environment by subjects under all test conditions (b). The use of DPV devices could raise the percentage of thermal acceptability at the three examined ambient temperatures. Especially when the devices were used at the flow rate set by the user, 100% of subjects reported that the thermal environment at 25 °C was acceptable, while only 10% said that the warmer thermal environments (i.e., ambient temperature at 27 °C and 29 °C) were unsatisfactory.

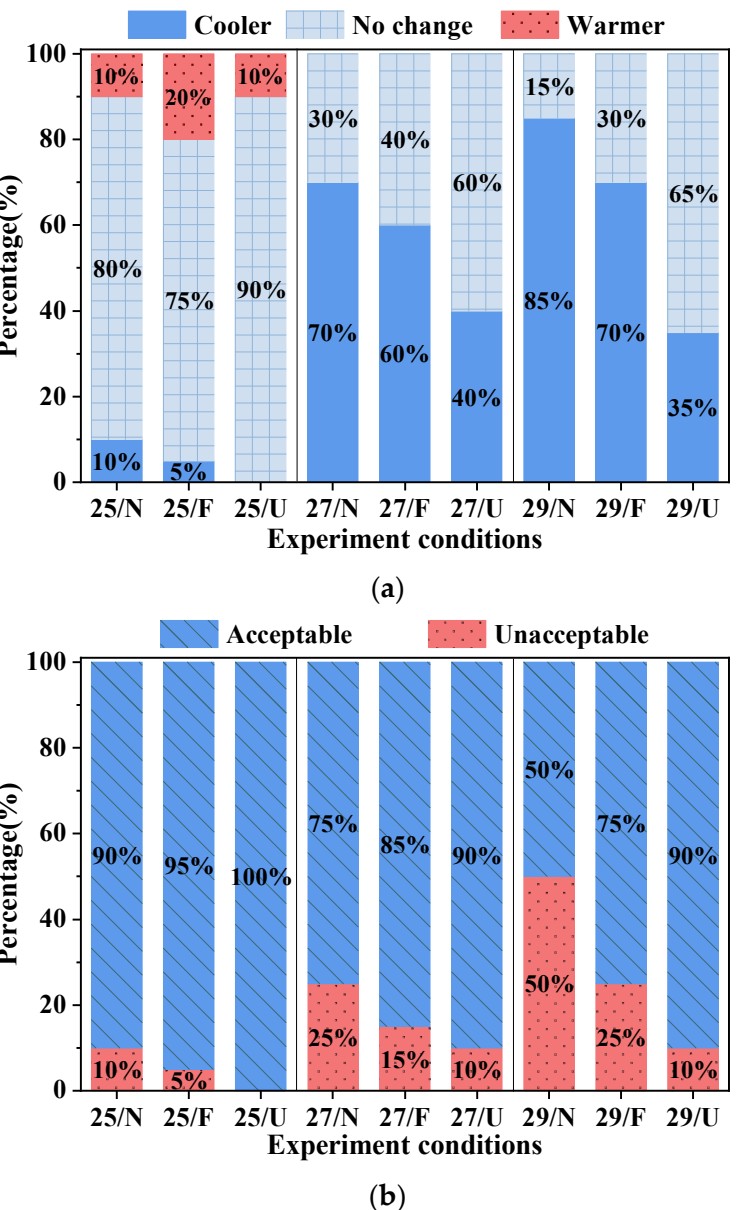

**Figure 8.** (**a**) Thermal preference; (**b**) Thermal acceptability.

It is worth noting that, while 70% of occupants chose no change at 25F, compared to 80% at 25N, their acceptability at 25F was higher. This is because thermal acceptability and thermal preference did not exactly correspond [30,42].

### 3.3.3. User Control Choices and Air Movement

Table 3 shows the subjects' choices for adjusting the flow rate of the devices at the end of stage 3. At 25 °C, 75% of the subjects chose to adjust the airflow rate. This because the pre-set flow rate was not set according to subjects' preferences. Individual differences in preferences are unavoidable [43]. As shown in Figure 5, more subjects turned down the flow, because the pre-set flow may be too high for them and caused them to feel cool. Although subjects reduced the flow at 25 °C, no one turned off the device. In fact, subjects see appropriate air movement as a positive influence in their work environment [44]. In warm environments, air movement has the potential to meet occupants' comfort [45]. More than 40% chose a higher DPV supply flow rate under warmer ambient conditions. Meanwhile, due to individual differences, 45% and 30% of participants, respectively, chose a lower flow rate.

Figure 9 shows the subjects' preferred air movement under all conditions. Under no-device conditions, more than 60% of subjects desired more air movement, i.e., higher air velocity surrounding their facial region. As a result, no subject was chosen to turn off the gadgets. After varying the flow rate of DPV devices based on their desire for air movement in the head region, 90% of participants indicated a willingness to "no alter" the air movement in stage 4. This demonstrated that a DPV flow of less than 10 L/s was suitable for subjects at 25 °C. The occupants' preference for air movement corresponded to their stage 3 choices.

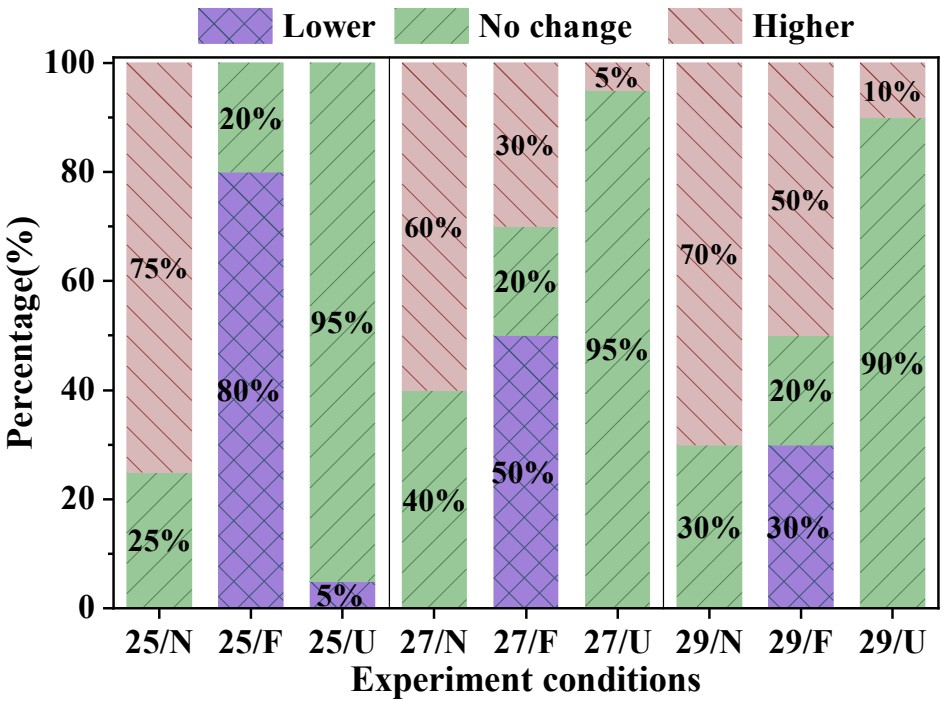

**Figure 9.** Occupants' preference for air movement in the face region.

### 3.4. Perceived Air Quality

Figure 10a depicts the acceptability of perceived air quality reported (PAQ) by participants. The use of DPV devices increased PAQ marginally at 25 °C indoor temperature, while the differences were not significant ($p = 0.052$ at fixed flow rate, $p = 0.218$ at user-control flow rate). When the indoor temperature was 27 °C, the PAQ could be greatly improved by using DPV devices with a set flow rate ($p < 0.001$). After users adjusted the DPV devices to their liking, the PAQ could still be slightly higher than in the absence of DPV. PAQ may be greatly improved at both flow rate modes when the DPV devices were utilized at 29 °C indoor temperature ($p < 0.001$).

Subjects reported considerably lower perceived inhaled air temperature (PAT) after using the DPV devices compared to no devices in all three ambient conditions (see Figure 10b). These differences were obviously significant at warmer ambient settings compared to neutral conditions. Such results obtained with an IJV DPV device could transport cooler air for occupants to breathe. Figure 10c depicts the relationship between perceived air temperature and perceived air quality, with the results revealing a substantial linear connection between PAT and PAQ ($R^2 = 0.92$). This finding has previously been reported in other studies [46].

Increased air movement would increase satisfaction with air quality in elevated room air temperatures with insufficient air conditions [33]. This explains why the difference in PAQ between with and without DPV flow was more pronounced in warmer conditions than in neutral conditions. Because the subject's flow rate selection was not detailed, only how they adjusted the knob was recorded; whether the air movement would influence the perceived air temperature was not discussed here.

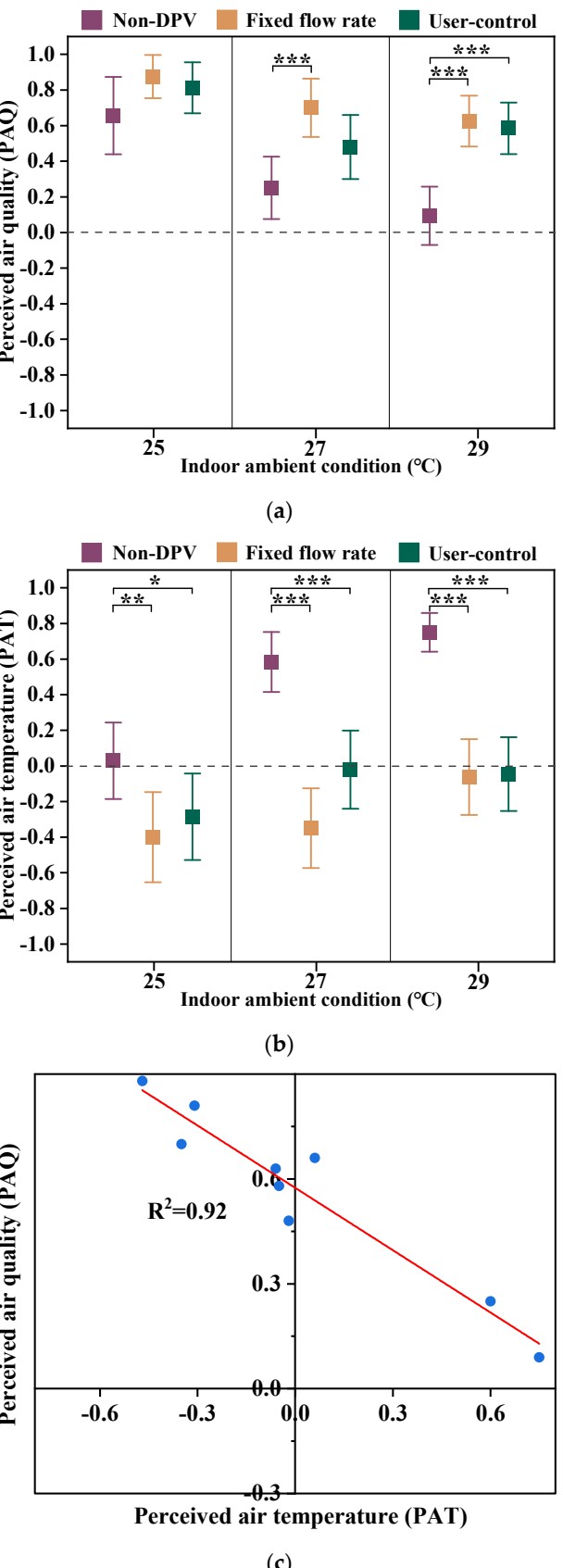

**Figure 10.** (**a**) Votes on perceived air quality; (**b**) Votes on perceived air temperature; (**c**) Relationship between perceived air quality and perceived air temperature. Note: for (**a**,**b**), mean value and 95% confidence interval (CI) are also shown. *, *p* < 0.05; ** *p* < 0.01; ***, *p* < 0.001.

### 3.5. Sick Building Syndromes

Figure 11a–d illustrate the response of several areas of the human body (nose, throat, face, and head) to different DPV conditions at varied indoor ambient temperatures. Subjects reported no irritation with the body parts when exposed to the examined thermal environment when no DPV devices were present, i.e., the DPV was turned off.

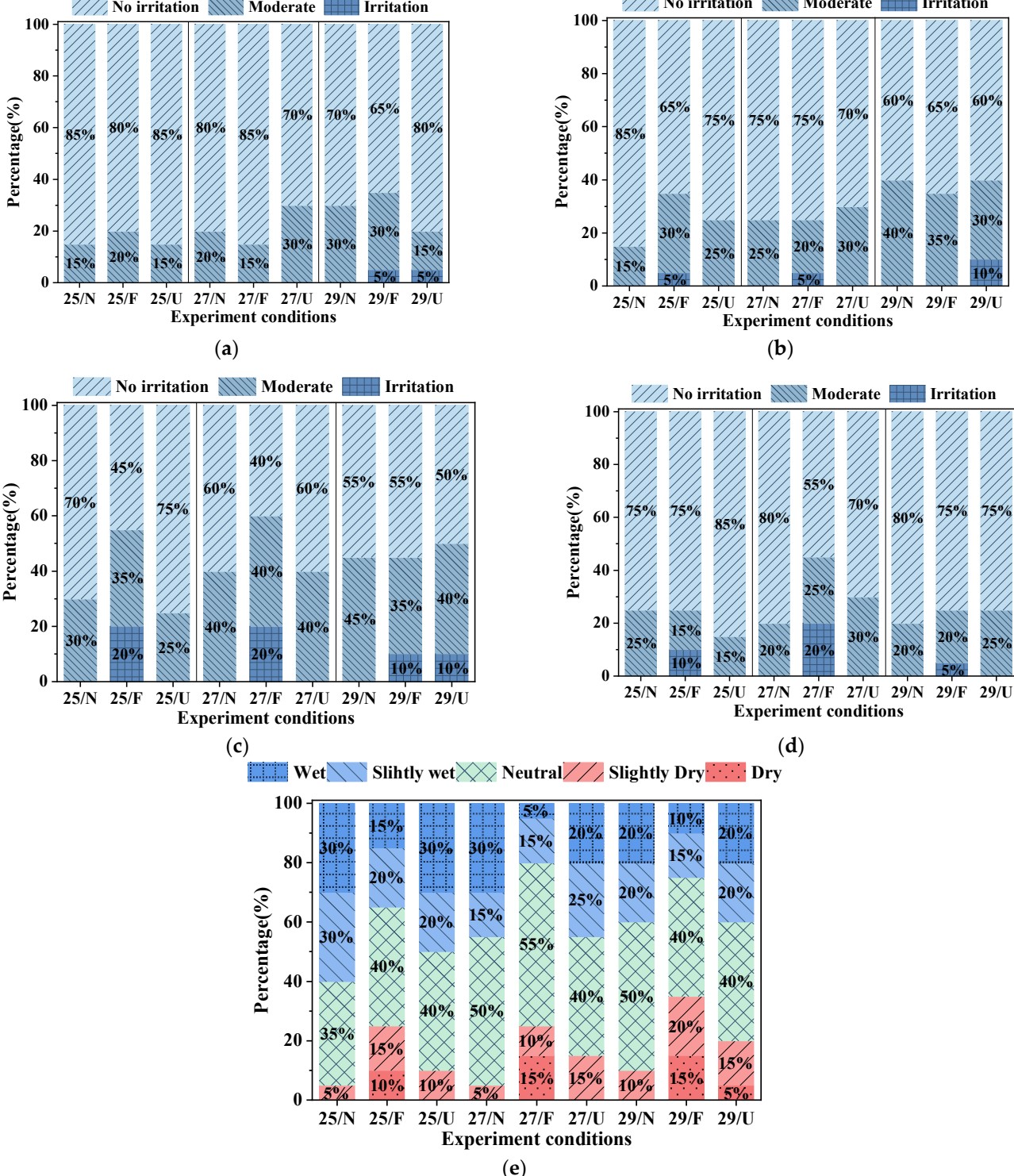

**Figure 11.** Response of different parts of subjects: (**a**) nose; (**b**) throat; (**c**) face; (**d**) head; (**e**) eyes.

No occupants reported nose discomfort when the DPV was run at the pre-set flow rate of 12 l/s at an indoor ambient temperature of 25 °C; however, 5%, 20%, and 10% of occupants experienced throat, face, and head irritations, respectively. At a temperature of 27 °C, 5%, 20%, and 20% of occupants reported throat, face, and head irritations, respectively. At 29 °C, 5%, 10%, and 5% of occupants reported nose, face, and head irritations, respectively. Because each occupant's preference for air movement differed, a pre-fix flow at the same level would not satisfy all subjects and might even irritate them. Except at 29 °C, no obvious irritation was reported when the device was run at the set flow rate according to subjects. This indicated that DPV with user control would be a better choice, but some irritations would be reported in order to provide more air movement for thermal comfort at 29 °C.

Dry eyes, the most common sign of eye irritation, should be carefully considered when operating a PV system [12,47]. The ratings of dry eyes are shown in Figure 11e. The findings revealed that using DPV devices at a fixed flow rate increased the percentage of occupants who reported dry eyes marginally. Subjects' dry eyes improved after they customized their DPV devices to their preferences. However, in warm conditions, such as 29 °C, dry eyes should be approached with caution due to irritation caused by high air movement [48].

*3.6. Energy Consumption*

Such an integrated system has the potential to improve thermal comfort and perceived air quality. As shown in Figure 8, even in the warm condition, i.e., 29 °C, 90% of subjects reported that the thermal environment was acceptable. As a result, such an integrated system saves energy. In addition, the energy consumption at each condition was calculated using Equation (1).

$$Q = cm\Delta T \tag{1}$$

$$\Delta T = T_O - T_S \tag{2}$$

where, $Q$ is the energy consumption for cooling, W; $c$ is the constant-pressure specific heat, $c$ = 1.005 kJ/(kg·°C); and m is the mass flow rate, which could be calculated by the suppl flow of IJV. *To* is the outdoor temperature of the Xi'an city in the summer, $T_O$ = 35 °C; *Ts* is the supply air temperature of IJV at each condition. Because no subjects chose to turn off the DPV device, the device's energy consumption was calculated using its maximum power, and the total energy consumption of the four devices was 56 W. It should be noted that this would overestimate the DPV devices' energy consumption.

As depicted in Figure 12, the total energy consumption in warm conditions with DPV devices for thermal comfort is significantly lower than in neutral conditions without DPV devices. When the thermal sensation is close to neutral, operating DPV devices at 27 °C saves 15.8% more energy than the neutral condition, i.e., at 25 °C. When at 29 °C, the energy savings approached 34%. This demonstrated that using DPV would improve thermal comfort and save energy. Meanwhile, irritation such as dry eyes that can occur at 29 degrees should be fully considered in practice.

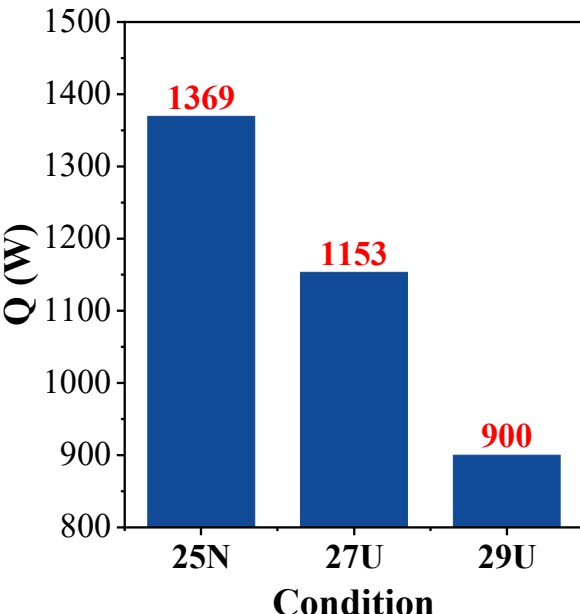

**Figure 12.** Comparison of total energy consumption between warm conditions with DPV devices and neutral condition without the DPV device.

## 4. Limitations and Futures Work

The results showed that combining IJV and DPV could improve thermal comfort and perceived air quality. Even in warm conditions, such an integrated system performed well, which would benefit energy savings. However, there are desirable targets that were not considered in this study, such as evaluating the work performance of subjects under such an integrated system to simplify the experiment and reduce the time cost. The specific air flow chosen by each subject was not counted; in fact, more detailed information about the subject's preferred airflow can be obtained, which is useful in practice.

Although the results show that the integrated system improved thermal comfort, the issue of dry eyes caused by increased flow should not be overlooked. Future research should concentrate on system operation and optimization under warm conditions, as this is beneficial to energy savings. Because of its high performance, the use of an integrated system in a large open office should be widely discussed.

## 5. Conclusions

The integrated system (DPV+IJV) was assessed in chamber testing, and human responses to exposure to such conditions were observed and analyzed. The results are summarized below.

The DPV flow could supply cool air to the subject's head zone, resulting in a 1.5 °C reduction in local skin temperature at the forehead. In addition, in warm conditions, subjects' local thermal sensation and overall thermal sensation could be reduced to near neutral levels. At the three temperatures studied, such an integrated system also improves thermal comfort, particularly under the user-controlled DPV flow condition; as a result, 90% of subjects reported that the thermal environment was acceptable to them. The perceived air quality improved as a result of such an integrated system. Benefits from the benefits, this integrated system could raise the acceptable HVAC temperature setpoint to 29 °C, resulting in an average energy savings of 34% over neutral conditions at 25 °C. Furthermore, for subjects at 25 °C, a DPV flow of less than 10 l/s was acceptable. The DPV user control mode is recommended. The integrated system is ultimately concluded to improve thermal comfort, perceived air quality, and individual air movement preference while saving energy. However, some SBS issues, such as dry eyes, should be noted under such an integrated system operating at 29 °C.

**Author Contributions:** Data curation, P.L., Y.L. and D.J.; Formal analysis, P.L. and Y.L.; Investigation, P.L., Y.L. and D.J.; Methodology, P.L. and F.W.; Supervision, B.Y. and F.W.; Writing—original draft, P.L.; Writing—review & editing, B.Y. and F.W.; Funding acquisition, B.Y. All authors have read and agreed to the published version of the manuscript.

**Funding:** This research was funded by National Natural Science Foundation of China, grant number No. 52278119.

**Institutional Review Board Statement:** The study was conducted in accordance with the Declaration of Helsinki, and approved by the Research Ethics Committee of Tianjin Chengjian University. Approval Code: TCU−20220410005 Approval Date: 10 April 2022.

**Informed Consent Statement:** Informed consent was obtained from all subjects involved in the study.

**Data Availability Statement:** The data presented in this study are available upon request from the corresponding author.

**Acknowledgments:** The authors would like to thank the participants who took part in this study.

**Conflicts of Interest:** The authors declare no conflict of interest.

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
