# Peer review of "Assessment of Thermal Comfort and Air Quality of Room Conditions by Impinging Jet Ventilation Integrated with Ductless Personalized Ventilation"

_sustainability, doi:10.3390/su141912526_

Round 1

Reviewer 1 Report

1. Add some concrete results from the experiment in abstract section.

2. Literature review is weak. Moe literature review of latest studies is required to establish the research gap.

3. Avoid using cumulative citations such as [15,16,20], [23,25,26]. Use of each citation must be explained individually.

4. Results and discussion must be combined as a single heading.

5. Kindly explain every parameter which is used to identify the indoor thermal comfort.

6. What is perceived air quality. How it affects the results?

7. Kindly add recommendation for future work and limitations of the current study.

Author Response

Reviewer #1:

[COMMENT]1. Add some concrete results from the experiment in abstract section.

[REPLY]

Thanks! The abstract has been modified, and some results have been added.

[COMMENT]2. Literature review is weak. More literature review of latest studies is required to establish the research gap.

[REPLY]

Thanks for the suggestions! The introduction has been modified, and more literature has been reviewed.

[COMMENT] 3. Avoid using cumulative citations such as [15,16,20], [23,25,26]. Use of each citation must be explained individually.

[REPLY]

Thanks for the suggestions! Such citations have been modified.

[COMMENT]4. Results and discussion must be combined as a single heading.

[REPLY] 

Thanks for the suggestions! Has been modified.

[COMMENT]5. Kindly explain every parameter which is used to identify the indoor thermal comfort.

[REPLY]

Apologized! The parameters related to thermal comfort, such as thermal comfort votes, thermal sensation votes, and thermal votes have been briefly explained in section 2.4. For example, .To obtain subjects' thermal states, the ASHRAE 7-point scale was used to assess thermal sensation (-3 cold, -2 cool, -1 slightly cool, 0 neutral, +1 slightly warm, +2 warm, +3 warm)[29]..

[COMMENT] 6. What is perceived air quality.  How it affects the results?

[REPLY]

Perceived air quality is an important factor in assessing the indoor environment, occupants evaluated the air quality as good when the air is perceived as fresh and pleasant [30]. And the perceived air quality index has been used in chamber tests for evaluating indoor air quality and has been the basis for current guidelines and standards for ventilation, e.g., EN 15251, and ASHRAE standard 62.2019.

So, the perceived air quality was used to evaluated the air quality in the present study. And it is explained in the revised manuscript.

Reference:

[30] A.K. Melikov, J. Kaczmarczyk, Air movement and perceived air quality, Build. Environ. 47 (2012) 400–409. https://doi.org/10.1016/j.buildenv.2011.06.017.

[COMMENT]7. Kindly add recommendation for future work and limitations of the current study.

[REPLY]

Thanks! The limitations of the current study and the future work have been added on Page 15.

Reviewer 2 Report

This study looks into employing ductless personalized ventilation in conjunction with impinging jet ventilation to improve thermal comfort and perceived air quality. The topic of the article is interesting. However, the significance and originality of the article are insignificant.

The following remarks must to be taken into account:

1)      The scope of the study should be declared clearly in the title, abstract and the introduction.

2)      The aim of the study is not clear. What about the ‘sick building symptoms’?

3)      The Introduction should be linked more preciously with the aim of the study. The introduction can be sharpened and linked more preciously with the aim of the study with the support of additional theoretical recent references that might be provided with specific regards to developing counties, socioeconomic factors, and building materials.

You did not provide related data to ‘thermal comfort’, ‘air quality’, and ‘sick building symptoms’. How did you link ‘dry eyes’ and ‘sick building symptoms’ with thermal comfort and air quality?

4)      The outline of the article needs restructuring.

You considered 3.3.4. Perceived air quality, 3.3.5. Dry eyes, and 3.3.6. Sick building syndromes, under the 3.3. Thermal comfort.

Why did you separate ‘dry eyes’ from Sick building syndromes?

5)      The stock charts (Figures 5, 6, 7, 10) are difficult to understand. Can you find an easier way to present the results?

6)      Figure 8 is not clear.

7)      The Results:

On line 206, you mentioned that ‘….the overall thermal sensation (OTS) was reduced. …’ and ‘ At 25 °C, the mean value dropped to -0.2, 0.5 at 27 °C, and 0.7 at 29 °C. …’. How does OTS reduced while the mean increased?

The same comment applies to the other sections of the results.

8)      The two parts (a) and (b) of Figure 6 should be presented at one page, or to be reorganized as landscape.

9)      How did you measure Sick building syndromes on pp.12? you need to introduce this at the beginning of the article. In addition, you need to link it more preciously with the aim of the study.

10)   The conclusion should be improved.

Add limitations of the study, are there any potentially relevant variables that you didn’t take into account?

Add recommendations for practical implications the system.

Author Response

Reviewer #2:

[COMMENT]1. The scope of the study should be declared clearly in the title, abstract and the introduction.

[REPLY]

Thanks! Has been modified.

[COMMENT]2. The aim of the study is not clear. What about the ‘sick building symptoms’?

[REPLY]

Apologized! Headache, runny nose, dry skin, throat, and dry eyes are collectively recognized as elements of the sick building syndrome. In order to evaluate the performance of the integrated system, an SBS questionnaire is necessary.

[COMMENT]3. The Introduction should be linked more preciously with the aim of the study. The introduction can be sharpened and linked more preciously with the aim of the study with the support of additional theoretical recent references that might be provided with specific regards to developing counties, socioeconomic factors, and building materials.

You did not provide related data to ‘thermal comfort’, ‘air quality’, and ‘sick building symptoms’. How did you link ‘dry eyes’ and ‘sick building symptoms’ with thermal comfort and air quality?

[REPLY]

Thanks for your kind suggestions! The introduction has been modified, and some recent references have been added to the revised manuscripts.

And as you suggested in comment 4, the outline has been restructured.

[COMMENT]4. The outline of the article needs restructuring. You considered 3.3.4. Perceived air quality, 3.3.5. Dry eyes, and 3.3.6. Sick building syndromes, under the 3.3. Thermal comfort. Why did you separate ‘dry eyes’ from Sick building syndromes?

[REPLY]

Thanks, and agreed that dry eye should be considered one of the sick-building syndromes.

And the outline has been restructured.

[COMMENT]5. The stock charts (Figures 5, 6, 7, 10) are difficult to understand. Can you find an easier way to present the results?

[REPLY]

Apologize for the confusion.

Here Figures 5, 6, and 7, are the boxplots, not the stock charts. Boxplots can describe the overall distribution of data using statistics such as mean (horizontal lines in the box), 25/% quantile (the lower boundary of the box), 75/% quantile (the upper boundary of the box), upper boundary (the upper whiskers), and lower boundary (the lower whiskers). So, here we used such charts to describe the data distribution.

Here Figure 10 are the interval graph, the middle graph is the average value, and the whiskers at both ends are error bars (90% confidence interval).

And explanations of the figures have been added.

[COMMENT]6. Figure 8 is not clear.

[REPLY]

Apologize and has been modified.

[COMMENT]7. The Results: On line 206, you mentioned that ‘….the overall thermal sensation (OTS) was reduced. …’ and ‘ At 25 °C, the mean value dropped to -0.2, 0.5 at 27 °C, and 0.7 at 29 °C. …’. How does OTS reduced while the mean increased? The same comment applies to the other sections of the results.

[REPLY]

 Apologized! I might describe not so clear in the manuscript.

Here, I want describe the OTS at all the studied conditions (25°C, 27°C, and 29°C) was decreased with the DPV flow comparing to no DPV flow.

The explain has been modified. …When DPV devices with a fixed flow rate were used in stage 2, the OTS was dramatically reduced. The mean value dropped from 0 to -0.2, 0.8 to 0.5 , and 1.1 to 0.7 at 25 °C, 27 °C, and 29 °C, respectively.  

[COMMENT]8. The two parts (a) and (b) of Figure 6 should be presented at one page, or to be reorganized as landscape.

[REPLY]

Apologize! And these have been modified (see page 6).

  [COMMENT]9. How did you measure Sick building syndromes on pp.12? you need to introduce this at the beginning of the article. In addition, you need to link it more preciously with the aim of the study.

[REPLY]

Thanks for the suggestions! In Section 2.4 of the revised manuscript, I have added descriptions of the SBS, and how to measure was also described in detail.

 [COMMENT]10. The conclusion should be improved. Add limitations of the study, are there any potentially relevant variables that you didn’t take into account? Add recommendations for practical implications the system.

[REPLY]

Thanks for the suggestions! Conclusions have been modified, and the limitations of the current study and future work have been added (see pages 15-16).

Reviewer 3 Report

Moderate revision is suggested. Please check the attachment for comments and questions.

Author Response

Reviewer #3:

 [COMMENT]1. If quantitative analysis has been conducted on the energy-saving performance of the targeted system, it’s suggested to add some quantified results on energy conservation to the end of the abstract.[REPLY]

Thanks!  The quantified results on energy have been added to the abstract.

 [COMMENT]2.  Line 47: “Even within the same person”, I assume it should be “even for one person”

[REPLY]

Modified, thanks!

 [COMMENT]3. Line 51: “allowing for”, I believe it should be “providing”

[REPLY]

Thanks, modified!

 [COMMENT]4. Figure 2(a), about the jet direction. Does the supply air flow from the nozzle, toward the red dot in the middle of the room? Please plot the ground air flow in Figure 2(a).

[REPLY]

Apologize for the confusion. Here the red dot was the position of the measurement instrument and Figure 2 has been modified.

Another question is: According to Figure 1, I thought the DPV equipment should be placed right above the ground air stream. But Figure 2 indicates that all DPV inlets in this experiment are pretty far from the ground air stream (if my assumption about the air stream direction is right). Is that going to influence the ventilation efficiency?

[REPLY]

Apologize for the confusion. Here Figure 1 just showed a schematic. In fact, the impinging jet is well known for high-performance air distribution, not only in the office room but also in large spaces, with high ventilation effectiveness. As shown in Figure 2(b), the DPV inlets are near the ground for intaking the air near the ground. And in our preliminary experiments, this layout has no negative effect.

 [COMMENT]5. Table 2 is confusing. If males' heights are within the range of 1.76±0.04,then why the heights of the whole group is within the range of 1.69±0.09?[REPLY]

Sorry for such confusion. The height distribution of the subjects was not uniform, and females are generally shorter than males in the experiment. So, this results were obtained by averaging over the whole group.

 [COMMENT]6. The heading of Section 2.5: “analysis”.

[REPLY]

Thanks, modified.

[COMMENT] 7. Section 2.3: I’m not quite familiar with the software “G*power”. Is this software designed to test if each volunteer could tell the difference of multiple temperatures? Moreover, it seems only 17 volunteers passed this test, and still, all 20 volunteers attended the experiment?

[REPLY]

Thanks. The “G*power” is software to help determine the number of sample sizes to participate in the experiment [31]. The calculated 17 means there should be at least 17 subjects participating in the present experiment to register a statistical difference of comparable magnitude.

In this study, all 20 volunteers participated in the experiment, which meets the required sample sizes of this study.

Reference:

[31] L. Lan, Z. Lian, Application of statistical power analysis–How to determine the right sample size in human health, comfort and productivity research, Build. Environ. 45 (2010) 1202–1213.

 [COMMENT] 8. Section 3.1: It’s suggested to add some qualitative discussion about why the forehead temperature should be monitored and analyzed. And what is the benefit of a decreased forehead temperature, to an office staff. This part of discussion could better reveal the advantage of the targeted system.

[REPLY]

Thanks! Related discussion has been added to explain the forehead temperature, and could be found in Section 3.1(see page 6)

Even though the face takes up a very small part of the body's surface area, face cooling could improve occupants’ thermal acceptability and thermal comfort[12,35]. PV system directly distributes cold air at a specific velocity to the breathing zone, lowering skin temperature at PV target areas[36,37]. …

 [COMMENT] 9. Section 3.2.1: As far as I’m concerned, higher OTS grade means a warmer sensation. So, at the 25-degree situation, why the OTS grade under user-defined flow, is higher than the OTS grade at no-device (no-DPV-flow) situation? Please add more discussion to explain this result. I assume the reason could be: the fixed-flow stage made the volunteers chilly, which influenced their thermal sensation judgement.

[REPLY]

Thanks for the suggestions. Agreed with your assumption, discussion has been added in Section 3.2.1.

 [COMMENT] 10. Section 3.2.2: Still, it’s suggested to add some qualitative discussion about why the local thermal sensation of head and neck is important for us to care about.

[REPLY]

Thanks for the suggestions! Related reference has been added and discussed in Section 3.2.2.

 [COMMENT] 11. I guess Figure 7 verifies my assumption in Comment 9. Volunteers felt chilly under the fixed-flow mode at 25 degrees.[REPLY]

Agreed.

 [COMMENT] 12. Figure 8, 25-degree situation: Figure 8(a) shows that 75% volunteers prefer “no change” under fixed-flow situation, 25 degrees. And 80% volunteers prefer “no change” under nodevice situation, 25 degrees. 80% is larger than 75%, it seems they prefer no-device over fixed-flow, at 25 degrees. While in Figure 8(b), the acceptability result is on the contrary.

This issue should be analyzed.[REPLY]

Thanks!  The issue has been discussed in 3.3.2 (see page 9).

[COMMENT] 13. Line 365-369: authors claimed “because of the increased perceived air quality provided by DPV devices, the percentage of occupants who were bothered by the environment was dramatically reduced.” But according to Figure 12, there’s no evident advantage of DPV over pure IJV situation (no-device scenarios). In many cases, no-device cases perform even better on the “no irritation” ratio. Authors need to be more objective when discussing about the experimental results.[REPLY]

Apologize! The discussion about the SBS has been added in Section 3.5 (see page 13).

 [COMMENT] 14. Since there’s no work about the energy efficiency of the targeted equipment in this study, it’s suggested to remove this part from the abstract.

[REPLY]

Thanks!  The quantified results on energy have been added to the abstract.

 [COMMENT] 15. Conclusions: It’s suggested to describe the pros and cons of the targeted equipment, not just its good side. Please emphasize its suitable and unsuitable working conditions based on the experimental results. For instance, leaving the flowrate for user themselves to define is usually a smart choice; this equipment is more suitable for 27-degree rooms rather than 25-degree rooms.[REPLY]

Thanks! The conclusions have been modified.

 [COMMENT]16. Finally, thank you all for introducing this new type of equipment in detail. I did learn something from this article.

[REPLY]

Thanks so much for all your suggestions!

Reviewer 4 Report

The manuscript presents a fascinating analysis of personalized ductless ventilation to create individual air circulation in an office scenario improving their thermal sensation. The experimental design and the parameters are well explained. The sample seems to be short, but the authors justify the size to be enough to detect a statistical difference. The references seem to be enough; nevertheless, it is recommendable to update some of the references or increase the number of recent references because only 27% are from 2017 to now, 44% are references between 2007-2016, and 2931% are older than 2006. Also, some of the leading global authors in thermal comfort are not mentioned enough, like Richard de Dear, Nichols, Nikolopoulos, or Indraganti. Also, the authors should clarify the reason for only monitoring the thermal environment at 1.1 m and not following the ISO 7726 ergonomics of the thermal environment.

Author Response

Reviewer #4:

The manuscript presents a fascinating analysis of personalized ductless ventilation to create individual air circulation in an office scenario improving their thermal sensation. The experimental design and the parameters are well explained. The sample seems to be short, but the authors justify the size to be enough to detect a statistical difference. The references seem to be enough; nevertheless, it is recommendable to update some of the references or increase the number of recent references because only 27% are from 2017 to now, 44% are references between 2007-2016, and 2931% are older than 2006. Also, some of the leading global authors in thermal comfort are not mentioned enough, like Richard de Dear, Nichols, Nikolopoulos, or Indraganti. Also, the authors should clarify the reason for only monitoring the thermal environment at 1.1 m and not following the ISO 7726 ergonomics of the thermal environment.

[REPLY]

Thanks for suggestions! Some recent related references have been added. And the work related to the present study of the global expert has been referenced.

Here the temperature at 1.1 m was monitored due to it’s the head level of seated person. Additionally, some related work also using such a method [15,30].

References:

[15] M. Dalewski, A.K. Melikov, M. Vesely, Performance of ductless personalized ventilation in conjunction with displacement ventilation: Physical environment and human response, Build. Environ. 81 (2014) 354–364.

[30] W. Su, B. Yang, B. Zhou, F. Wang, A. Li, A novel convection and radiation combined terminal device: Its impact on occupant thermal comfort and cognitive performance in winter indoor environments, Energy Build. 246 (2021) 111123.

Round 2

Reviewer 1 Report

Accepted

Reviewer 2 Report

The new version of the paper addressed the concerns raised in the first review and the overall quality has been improved.